# Potential Therapeutic Role of Phytochemicals to Mitigate Mitochondrial Dysfunctions in Alzheimer’s Disease

**DOI:** 10.3390/antiox10010023

**Published:** 2020-12-28

**Authors:** Md. Ataur Rahman, MD. Hasanur Rahman, Partha Biswas, Md. Shahadat Hossain, Rokibul Islam, Md. Abdul Hannan, Md Jamal Uddin, Hyewhon Rhim

**Affiliations:** 1Center for Neuroscience, Korea Institute of Science and Technology (KIST), Seoul 02792, Korea; 2Global Biotechnology & Biomedical Research Network (GBBRN), Department of Biotechnology and Genetic Engineering, Faculty of Biological Sciences, Islamic University, Kushtia 7003, Bangladesh; 3ABEx Bio-Research Center, East Azampur, Dhaka 1230, Bangladesh; hasanurrahman.bge@gmail.com (M.H.R.); parthabiswas2025@gmail.com (P.B.); shahadat4099@gmail.com (M.S.H.); hasan800920@gmail.com (M.J.U.); 4Department of Biotechnology and Genetic Engineering, Bangabandhu Sheikh Mujibur Rahman Science and Technology University, Gopalganj 8100, Bangladesh; 5Department of Genetic Engineering and Biotechnology, Jashore University of Science and Technology, Jashore 7408, Bangladesh; 6Department of Biotechnology and Genetic Engineering, Noakhali Science and Technology University, Noakhali 3814, Bangladesh; 7Department of Biotechnology and Genetic Engineering, Faculty of Biological Sciences, Islamic University, Kushtia 7003, Bangladesh; rakibbgeiu@yahoo.com; 8Department of Biochemistry, College of Medicine, Hallym University, Chuncheon, Gangwon-do 24252, Korea; 9Department of Anatomy, Dongguk University College of Medicine, Gyeongju 38066, Korea; hannanbmb@bau.edu.bd; 10Department of Biochemistry and Molecular Biology, Bangladesh Agricultural University, Mymensingh 2202, Bangladesh; 11Graduate School of Pharmaceutical Sciences, College of Pharmacy, Ewha Womans University, Seoul 03760, Korea; 12Division of Bio-Medical Science and Technology, KIST School, Korea University of Science and Technology (UST), Seoul 02792, Korea

**Keywords:** Alzheimer’s disease, mitochondrial dysfunctions, phytochemicals, reactive oxygen species (ROS), autophagy

## Abstract

Alzheimer’s disease (AD) is a progressive neurodegenerative disorder characterized by a decline in cognitive function and neuronal damage. Although the precise pathobiology of AD remains elusive, accumulating evidence suggests that mitochondrial dysfunction is one of the underlying causes of AD. Mutations in mitochondrial or nuclear DNA that encode mitochondrial components may cause mitochondrial dysfunction. In particular, the dysfunction of electron transport chain complexes, along with the interactions of mitochondrial pathological proteins are associated with mitochondrial dysfunction in AD. Mitochondrial dysfunction causes an imbalance in the production of reactive oxygen species, leading to oxidative stress (OS) and vice versa. Neuroinflammation is another potential contributory factor that induces mitochondrial dysfunction. Phytochemicals or other natural compounds have the potential to scavenge oxygen free radicals and enhance cellular antioxidant defense systems, thereby protecting against OS-mediated cellular damage. Phytochemicals can also modulate other cellular processes, including autophagy and mitochondrial biogenesis. Therefore, pharmacological intervention via neuroprotective phytochemicals can be a potential strategy to combat mitochondrial dysfunction as well as AD. This review focuses on the role of phytochemicals in mitigating mitochondrial dysfunction in the pathogenesis of AD.

## 1. Introduction

Several studies have demonstrated that mitochondrial dysfunction leads to several neurodegenerative diseases, including Alzheimer’s disease (AD) [1,2,3]. AD shows common symptoms such as insanity and leads to a morbid state and death in the aged population [4]. In both familial and sporadic patterns, AD is characterized by dual unique medical hallmarks: senile plaques formed via the extracellular accumulation of amyloid-β (Aβ) peptide and intracellular deposition of neurofibrillary tangles (NFTs) formed via hyperphosphorylation of tau proteins [5,6]. These phenomena are accompanied by both pre- and postsynaptic and neuronal casualty [4,7]; however, the pathogenesis of AD pathogenesis is still unclear. In addition, multiple reports demonstrate that the alterations in axonal transport (AT) are the precise culprit for the development of neurodevelopmental diseases such as AD [8]. AD in mammals involves the atypical decomposition of several abnormal organelles like mitochondria, resulting in the degeneration of senile plaques along with abnormal neuronal expansion leading to a decline in neurites [9]. Phytochemicals or plant-derived chemical compounds are currently under research with unestablished health benefits [10]. Phytochemicals show multiple beneficial effects on mitochondrial dysfunction [11]; however, enough investigations have not been performed yet examining their clinical application. 

A wide range of studies have demonstrated that numerous bioactive phytochemicals and other organic compounds may improve the treatment of AD [12]. Phytochemicals, including polyphenolic compounds that are present in numerous plants exhibit several essential properties such as anti-inflammatory potential, DNA repair, autophagy, and antioxidant activities [13]. In the brains of AD patients as well as transgenic AD mouse models, APP and Aβ are present in mitochondrial membranes, interrupting the mitochondrial electron transport system [14]. Potential therapeutic effects of these phytochemicals include antioxidant and anti-inflammatory activities via modulation of Aβ toxicity. Mitochondrial dysfunction discharges excessive quantities of H_2_O_2_, which ultimately leads to irreversible cellular dysfunction and damage in the brain [15]. Aggregated Aβ peptides, H_2_O_2_-induced hydroxyl radical, and mitochondrial dysfunction caused by APP in AD may restrain in addition to pharmacological approaches using phytochemicals that preserve mitochondrial dynamics [16]. Owing to their therapeutic capabilities, phytobioactive compounds have been deliberated as favorable beneficial agents for AD and age-related diseases [17]. Therefore, the current review describes the underlying mechanisms of mitochondrial dysfunction in the pathogenesis of AD and discuss how phytochemicals may mitigate mitochondrial dysfunction.

## 2. Mitochondrial Dysfunction in AD via ROS Production

Oxidative stress (OS) occurs owing to the imbalance between the generation of reactive oxygen species (ROS) and cellular antioxidant potential. OS stands for excess quantities of ROS production that incur damage to nucleic acids and small molecules such as proteins or lipids. OS can lead to neuronal, specifically causing neurodegenerative diseases and cellular aging processes [18]. Restrained ROS production has physiological roles, particularly in controlling cellular redox equilibrium and regulating intracellular signal transduction [19,20]. ROS (collectively, H_2_O_2_, OH, and O_2_)^·−^) may be the causative factor leading to defects in mitochondrial respiration and the development processes of the human brain that are accompanied by augmented ROS generation. They also contribute to dynamic changes in the brain during AD and aging progression (Figure 1).

The primary origins of ROS production in the brain under functional circumstances as well as in pathological processes (e.g., neurological diseases) are complex I and complex III of the respiratory chain. Complex I discharge superoxide (O_2_^·−^) into the intermembrane space such as the matrix, and complex III liberates O_2_^·−^ to both sides of the electron transport chain (ETC) or inner mitochondrial membrane. Hydrogen peroxide (H_2_O_2_) can be generated from O_2_^·−^ by an enzyme called superoxide dismutase. Both molecules can cross the inner membranes and can produce extremely reactive hydroxyl radical (^·^OH). Under physiological conditions, the proton movements and the respiratory state of mitochondria produce H_2_O_2_ and O_2_^·−^ from the electron transport chain (ETC) [21]. Complex IV also enhances the generation of ROS, whereas complexes III and V generate a minimal amount of ROS [22]. Apart from these, defective production and detoxification of ROS are critically involved in mitochondrial dysfunction [23]. During the aging process, a high amount of ROS is generated due to defective mitochondria. Likewise, a decline in antioxidant enzyme activities ensues, leading to increased ROS production [23,24]. Excess ROS production adversely affects the ETC; complexes I, III, and IV appear to be the most affected, while complex II remains undisturbed [23,25].

## 3. Mitochondrial Deformity as an Outcome of AD Progression

Accumulating evidence has demonstrated that metabolic alterations play a pivotal role in AD progression mediated by several pathogenic factors such as ROS, mitochondrial deformity, and Aβ load [26]. Extensive research has shown that ROS formation mediated by Aβ and calcium imbalance leads to mitochondrial injuries (Figure 2), which are categorized as a secondary mitochondrial failure. Hippocampal expression of mutant APP and Aβ in mouse HT22 cell lines led to impaired mitochondrial dynamics, alterations of mitochondrial structure, and action in neurons [27]. Amyloid precursor proteins (APP) can lead to the overexpression of mitochondrial protein import channels in AD sensitive brain regions, leading to mitochondrial malfunction [28]. Alternatively, several studies have shown that Aβ precisely disorganizes mitochondrial dynamics and hinders critical enzymatic functions. Lustbader et al. reported that Aβ-binding alcohol dehydrogenase (ABAD) directly interacts with Aβ and leads to Aβ-linked apoptosis, mitochondrial toxicity, and free-radical formation in neuronal cells [29]. Furthermore, voltage-dependent anion-selective channel 1 protein (VDAC1) is excessively expressed in AD-vulnerable brains, which combines with phosphorylated tau and Aβ to block mitochondrial intramembranous pores, accelerating mitochondrial impairment [30]. A distinct number of in vitro analysis proposed a connection among augmented Aβ levels, mitochondrial abnormal function, and oxidative burden, collectively leading to AD progression. Nevertheless, the origin of the impairment of mitochondrial dynamics in AD pathogenesis remains elusive.

## 4. Phytochemicals Prevent Mitochondrial Dysfunction and Improve Biogenesis

Several phytochemicals function to neutralize ROS and activate cellular antioxidant mechanisms. Phytochemicals also enhance mitochondrial biogenesis and protect neurons from toxic damage [31]. Additionally, phytochemicals can stimulate cell survival pathways by triggering many growths signaling pathways. In this section, we discuss recently explored phytochemicals that have been shown to protect neurons from mitochondrial dysfunction in AD by stimulating numerous signaling pathways. Molecular targets, experimental model, research outcomes, and molecular signaling systems of these phytochemicals are summarized in Table 1. Additionally, epidemiological as well as clinical interventions have been displayed that dietary phytochemicals, for example Mediterranean diet, exhibit beneficial properties in dementia patients, AD, PD, depressive disorders, and mild cognitive impairment [32]. Secondary metabolites of phytochemicals from Mediterranean diet contain ω-3 polyunsaturated fatty acids which has been described to maintain cognitive function in human studies [33]. 

Anthocyanins control mitochondrial fission/fusion pathways, prevent complex I APP Swedish K670N/M671L double mutation (APPswe), and promote normal mitochondrial dynamics [34]. Numerous phenolic compounds exert neuroprotective effects in AD and other neurodegenerative diseases and. Sulfuretin, a well-known flavonoid glycoside derived from *Albizia julibrissin*, protects primary hippocampal neuronal cells and SH-SY5Y neuroblastoma cells from Aβ-mediated neurotoxicity [35]. Dietary (poly)phenols have been found to cross blood-brain barrier (BBB) in endothelial cells and shown neuroprotective potential [36]. Polyphenol resveratrol, derived from grapes and black barriers, protects HT22 and PC12 cells against Aβ toxicity by activating the PI3K/Akt/Nrf2 pathway [37]. In addition, resveratrol prevents cell death and represses ROS production induced by Aβ toxicity by enhancing PI3K/Akt phosphorylation, the protein levels of SOD, HO-1, and GSH, and Nrf2 nuclear translocation [38]. Resveratrol also found to cross BBB [39]. Quercetin, a hydroxytyrosol derived from olives, prompts mitochondrial biogenesis and enhances muscle mtDNA in adult men [40]. Tea polyphenols (TPs) mitigate OS in H_2_O_2_-induced human neuroblastoma SH-SY5Y cells via Keap1-Nrf2 signaling initiation and decrease in H_2_O_2_-mediated cell death, as well as ROS and H_2_O_2_ levels to protect against mitochondrial dysfunction [41]. Liquiritigenin prompts mitochondrial fusion and prevents mitochondrial cytotoxicity, in addition to the fragmentation induced by Aβ in SK-N-MC cells [42]. In addition, EGCG and resveratrol increase the levels of Sirt-1 and AMPK along with mitochondrial biogenesis via PGC-1α, thereby protecting the neuronal cells [43]. Conversely, kaempferol, resveratrol luteolin, wogonin, quercetin, theaflavins, EGCG, curcumin, and baicalein open the mPTP, which activates the apoptosis pathway in cancer cells via Bcl-2 and Bcl-xL inhibition along with oligomerization of Bax, in addition to the downregulation of NF-κB signaling pathway [44]. Additionally, curcumin has been found to cross BBB to enter brain tissue and considerable exhibited neuro-protective as well as anti-cancer properties [45]. A previous study showed that curcumin protected against mitochondrial degeneration by mitigating the autophagic pathway via modulation of the PI3K/Akt/mTOR pathway in the ischemia/reperfusion-induced rat model [46]. Kaempferol passed to penetrate BBB and attenuates neuroinflammation as well as BBB dysfunction in cerebral ischemia/reperfusion rats in addition to improve neurological deficits [47]. A ginseng-derived exogenous lysophosphatidic acid receptor ligand, gintonin, improves blood-brain barrier permeability in primary human brain microvascular endothelial cells (HBMECs) [48].

### 4.1. Phytochemical Intervention of Molecular Signaling Pathways Related to Mitochondrial Dysfunctions in AD

Accumulating evidence has indicated that a large number of phytochemicals are capable of showing numerous benefits against mitochondrial dysfunction in AD pathogenesis by modulating molecular signaling pathways. Several polyphenols promote mitochondrial functions and biogenesis, particularly by regulating ETC activity, redox state modulation, and apoptosis inhibition. Phenolic acids can scavenge peroxynitrite, superoxide, and hydroxyl radicals, terminate radical chain reactions, and upregulate several protective genes that encode extracellular signal-related kinase 1/2 (ERK1/2), heat shock protein 70, and heme oxygenase-1 (HO-1) [11]. Several in vivo and in vitro studies have revealed that curcumin can prevent mitochondrial dysfunction in AD by scavenging hydroxyl radicals, hydrogen peroxide, and peroxynitrite and attenuating lipid peroxidation [55]. Flavonoids exhibit antioxidant activity and protect neurons via modulation of cellular signaling pathways, in addition to the induction of expression of several genes [56]. Flavonoids can also increase the expression of ROS-eliminating enzymes such as catalase, SOD, and glutathione reductase via the activation of the Keap1/Nrf2/ARE-mediated signaling pathway [57]. Polyphenols such as catechin, apigenin, luteolin, kaempferol, curcumin, and quercetin can inhibit ROS-generating xanthine oxidase (XO), NADPH oxidase (NOX), and MAO [58,59].

Flavonoids exert neuronal effects via several lipid kinase and protein kinase signaling pathways, such as the protein kinase C, MAPK tyrosine kinase, PI3K/Akt signaling pathways, and NF-κB pathway [60]. The stimulatory or inhibitory properties of these pathways can significantly modulate gene expression by altering the phosphorylation state as well as affecting the neuronal properties and function of target molecules. As a result, this might lead to synaptic protein synthesis, morphological variations, and plasticity involved in neurodegenerative processes in AD. Serine/threonine kinases, known as MAPK and mitogen-activated kinases, regulate numerous cellular functions via extracellular signal transduction pathways, generating intracellular downstream signals [61]. Flavonoids have selectively interacted with MAPK kinases, including ERK, MEK1, and MEK2 signaling, resulting in the activation of downstream cAMP response element-binding protein (CREB) [62]. These results might lead to alterations in memory function and synaptic plasticity via the upregulation of neuroprotective pathways in AD. 

Blueberry supplementation rich in anthocyanins and flavonols increased memory performance in rats via CREB activation and promoting pro-BDNF and mature BDNF levels in the hippocampus [63]. In another study, 12 weeks of blueberry supplementation activated Akt phosphorylation, mTOR downstream activation, and enhanced activity-regulated cytoskeletal-associated protein (Arc/Arg3.1) expression in the hippocampus of aged animals [63]. This might promote morphology and spine density in neuronal cells, thereby enhancing learning and memory function. In addition, treatment with green tea catechins ameliorated memory impairments and promoted spatial learning function by diminishing the oligomers of Aβ (1–42) in senescence-accelerated mice by augmenting the expression of the PKA/CREB pathway in the hippocampus [64]. Furthermore, EGCG promoted ERK and PI3K-mediated phosphorylation of CREB as well as stimulated GluR2 levels and modulated synaptogenesis, neurotransmission activity, and plasticity in cortical neurons [65]. In addition, flavonoids modulate the activity of PI3K via direct interactions with its ATP binding site [66]. Hesperetin is an activator of the Akt/PKB pathway in cortical neurons. In contrast, quercetin inhibits the prosurvival Akt/PKB pathways by preventing the activity of PI3K [67]. 

Flavonoids prevent certain activities of CDK5/p25 and GSK-3β, which contribute to the hyperphosphorylation of Tau and accumulation of neurofibrillary tangles in AD pathogenesis [62]. Indirubins prevent CDK5/p25 and GSK-3β and inhibit abnormal phosphorylation of tau in AD pathogenesis [68]. Likewise, GSK-3β activity is inhibited by flavonoid morin [69]. Morin can prevent GSK-3β-mediated phosphorylation of tau in vitro, decrease Aβ-induced tau phosphorylation, and protect against Aβ cytotoxicity in human SH-SY5Y neuroblastoma cells [69]. Furthermore, morin reduces the hyperphosphorylation of tau in the hippocampal neurons of 3xTg-AD mice [69]. Luteolin reduces soluble Aβ, interrupted the PS1-APP association, and diminished GSK-3 activity in an AD mouse model of Tg2576, and rescued cognitive impairments [70]. 

### 4.2. Phytochemicals Inhibit AD Specific Protein Aggregation

Neuropathological characteristics of AD involve the accumulation of amyloid-β plaques, neurofibrillary tangles, and neuronal loss in the limbic neocortical brain regions [71]. Pathobiology of AD encompasses oxidative stress, mitochondrial dysfunction, neuroinflammation, apoptosis, reduced neurotrophic factors and neurogenesis, loss of the cholinergic system, autophagy dysfunction, and glutamatergic excitotoxicity [72,73]. Various phytochemicals, anti-inflammatory medications, and antioxidants prevent amyloidogenic monomer synthesis, fibrillar aggregates, and oligomeric formation [74]. Phytochemicals also stimulate nontoxic aggregate formation and proteolytic system activation to ameliorate neuronal mitochondrial dysfunction triggered by Aβ [75]. It is well known that amyloidogenic Aβ 40–42 is produced via consecutive APP cleavage mediated by β-secretase (BACE1) and γ-secretase enzymes [76]. Tannic acid, genistein, ferulic acid, nobiletin, galangin, sinensetin, and tangeretin inhibit β-secretase activity, in addition to behavioral enhancement in AD animal models [11]. In addition, resveratrol, EGCG, icariin, quercetin, luteolin, 7,8-dihydroxyflavine, rutin, and curcumin decrease β-secretase expression and protect neurons [77]. Furthermore, curcumin, oleuropein, genistein, and EGCG promote APP cleavage via α-secretase, producing nontoxic N-terminal soluble APPα product and C-terminal α fragment [78]. Phytochemicals promote α-secretase or prevent β-secretase activity and inhibit fibril and toxic oligomer production [67]. Curcumin as well as other polyphenolic compounds have been changed to mature Aβ aggregates, which make nontoxic molecules as well. 

Many phytochemicals inhibit mTOR signaling, thereby inducing the autophagy pathway [6,79]. Polyphenols inhibit oligomer synthesis and formation, in addition to preventing tau hyperphosphorylation and aggregation reduction under in vitro and in vivo conditions [80]. Soluble Aβ oligomers along with profibrillar species are produced via the action of rosmarinic acid, myricetin, and curcumin, which reduce the number of toxic oligomers and fibrils [81,82]. Aβ aggregation is inhibited by honokiol, myricetin, and luteolin upon binding to the hydrophobic site of the amyloid pentamer and employed the most prominent Aβ1-42 aggregate inhibition in PC12 cells to protect anti-aggregative properties as well as neuronal toxicity [83]. Numerous phytochemicals involved in the pathogenesis of AD are indicated in Figure 3. Another potential benefit of phytochemicals in AD may include their potential role in tau phosphorylation. Tau oligomers are toxic and cause synaptic dysfunction in AD. Several findings have revealed that hyperphosphorylation of tau can be inhibited by treatment with caffeic acid, altenusin, EGCG, curcumin, and resveratrol [84,85]. Moreover, EGCG inhibits the conversion of tau aggregates into toxic oligomers [86]. In addition, emodin and daunorubicin repress tau aggregation and dissolve paired helical filaments under in vitro conditions [87]. In another study, epicatechin-5-gallate and myrecetin were shown to hinder heparin-mediated tau formation, and EGCG administration in an AD transgenic mouse model controlled the phosphorylation of sarkosyl-soluble tau isoform [88,89].

## 5. Therapeutic Applications of Phytochemicals in Mitochondrial Dysfunction in AD

Many studies have reported the effective therapeutic potential of antioxidants and mitochondria-targeting agents such as vitamin C, vitamin E, carnitine, and alpha-lipoic acid in AD [90]. The coenzyme Q10, piracetam, simvastatin, curcumin, ginkgo biloba, piracetam, and omega-3 polyunsaturated fatty acids also show effective therapeutic potential [91]. An effective therapeutic strategy can be developed against AD by targeting mitochondrial proteins. Using these strategies, various types of mitochondria-targeted antioxidants have been manufactured. The alteration of mitochondrial movement has a negative impact on mitochondrial function, thereby contributing critically to the pathogenesis of AD [92]. Consequently, approaches to modify defective mitochondrial movement and transportation may constitute an effective therapeutic strategy for the treatment of AD. Therapeutics that decrease the activation of the mitochondrial fission proteins such as Drp1, pTau, and Aβ can rescue the neurons from the toxic effects of those agents and their interconnection. A diversity of phytochemicals available in numerous plant sources demonstrate various pharmacological properties, including neuroprotection [93,94], apoptosis induction [95,96,97,98,99,100,101,102], autophagy activation [79,103,104,105], antioxidant [106] and anti-inflammatory action [107], and DNA repair function [13]. Because of these capabilities, phytochemicals are progressively considered as favorable therapeutic candidates for AD therapy [10] (Figure 4).

The therapeutic possibilities of curcumin were considered in various aging-related pathological disorders, including type 2 diabetes, ocular diseases, cancer, atherosclerosis, osteoporosis, rheumatoid arthritis, chronic kidney disorders, hypertension, cardiovascular diseases, and neurodegenerative disorders [108]. The neuroprotective action of curcumin in AD is well known. Curcumin protects against Aβ-mediated mitochondrial dysfunction and synaptic toxicity in SH-SY5Y human neuroblastoma cells [109]. However, the effects of curcumin in placebo-controlled, double-blinded clinical trials with AD patients were moderately inadequate [110]. Low solubility might be a potential cause. Recently, several preclinical investigations have claimed the anti-AD potential of quercetin [111]. Treatment with quercetin improved mitochondrial dysfunction by restoring mitochondrial membrane potential, which led to reduced ROS production, in addition to restoring ATP synthesis [112]. Furthermore, this treatment significantly enhanced AMPK expression, decreased scattered senile plaque formation, and inhibited the impairment of learning and memory [112]. More recently, in triple transgenic AD mice, the long-term oral administration of quercetin led to reduced tauopathy, astrogliosis, microgliosis, and β-amyloidosis in the amygdala and hippocampus, which improved cognitive function and performance of learning and spatial memory function [113,114]. Different phytochemicals and other chemicals used in mitochondrial-targeted AD treatments in preclinical and clinical studies are listed in Table 2. ω-3 fatty acid derived from fish consumption prevents coronary artery disease, stroke, aging, dementia, and AD is addressed in human trails [33]. However, in human, flavonoids and polyphenols from Mediterranean diets have been shown antioxidant as well as anti-inflammatory activities in cardiovascular disease, type-2 diabetes mellitus, cancer prevention, and stroke [33]. Fruits and vegetables which contains polyphenols have been modulated hyperphosphorylation of tau and Aβ aggregation in animal models of AD [33].

Genistein, a soy isoflavonoid, has therapeutic implications in many aging-related mitochondrial dysfunctions in pathological conditions, including neuroinflammation, oxidative stress, and aggregation of Aβ in AD. This therapeutic effect of genistein was attributed to its ability to improve function impairments induced by Aβ aggregates in mitochondrial dysfunctions [115]. However, genistein pretreatment in a primary astrocyte culture prevented Aβ-mediated production of pro-inflammatory mediators [116]. Recently, in a streptozotocin-induced rat model, a higher dose of genistein (150 mg/kg/day) was found to activate autophagy in the sporadic form of AD [51]. Additionally, genistein treatment resulted in the complete degradation of tau hyperphosphorylation and Aβ proteins in the brain of mitochondrial dysfunction. Currently, it has been innovated the designs in nanocomposites with genistein-loaded which has confirmed to develop the oral delivery system in addition to overcome the toxic effects isoflavonoid [117].

Plant polyphenols stimulate mitochondrial biogenesis and diminish mitochondrial dysfunction in AD [139]. Resveratrol represses the activity of cAMP phosphodiesterases and augments cAMP via the cAMP/CaMK/AMPA activation pathway [140]. Additionally, mitochondrial dynamics, biogenesis, and function have been activated by resveratrol via the activation of AMPK, protein kinase C epsilon (PKCε), along with the improvement in NAD^+^ levels [141]. In contrast, EGCG promoted the biogenesis of mitochondrial function in the AD model with Down’s syndrome through the Sirt1/PGC-1α signaling pathway via the upregulation of TFAM and Nrf1, in addition to mtDNA content [142]. Several flavones such as wogonin, quercetin, and baicalein improved the biogenesis of mitochondrial activities by enhancing the expression of Sirt1/AMPA/PGC-1α under in vitro and in vivo conditions [143]. Extra virgin olive oil contains oleuropein, augments mtDNA along with the expression of PGC-1α, complex II and IV, controlled mitogenesis and mitochondrial biogenesis in AD, and diminishes oxidative stress [144]. Therefore, pharmacological intervention via polyphenols has been anticipated as a promising therapeutic approach for mitochondrial dysfunction-associated neurodegenerative disorders.

## 6. Concluding Remarks and Future Directions

Although the prevalence of AD is increasing tremendously, there is still no specific therapeutic strategy for curing, slowing the progression, or prevention of AD [145]. Mitochondrial dysfunction plays a crucial role in the pathogenesis of AD. However, the elusive mechanisms of AD pathobiology further complicate treatment strategies. From this perspective, ongoing research is dedicated to underscoring the precise pathomechanism of AD as well as exploring the possibility of alternative treatment strategies. In light of the current discussion, pharmacological intervention via natural products, particularly phytochemicals, is a promising strategy to combat AD-associated pathological factors, including mitochondrial dysfunction. Phytochemicals and other natural compounds can prevent mitochondrial dysfunction by regulating several signaling pathways, including those associated with cellular antioxidant defense, anti-inflammation, autophagy and other quality control systems, mitochondrial biogenesis, and cell survival. Although several phytochemicals have shown promise against AD, their clinical application remains elusive. Since the therapeutic applications of many phytochemicals are limited owing to their poor pharmacokinetic properties, strategies such as nanoparticle synthesis may potentially improve their drug-likeness. Moreover, not enough clinical evidence is available compared to the preclinical data. Therefore, further human trials are necessary to translate the existing findings into clinical use. Understanding the advanced pathobiology of AD and the pharmacological mechanism of phytochemical-based therapy may offer an emerging novel neuroprotective approach for AD in the future.

## Figures and Tables

**Figure 1 antioxidants-10-00023-f001:**
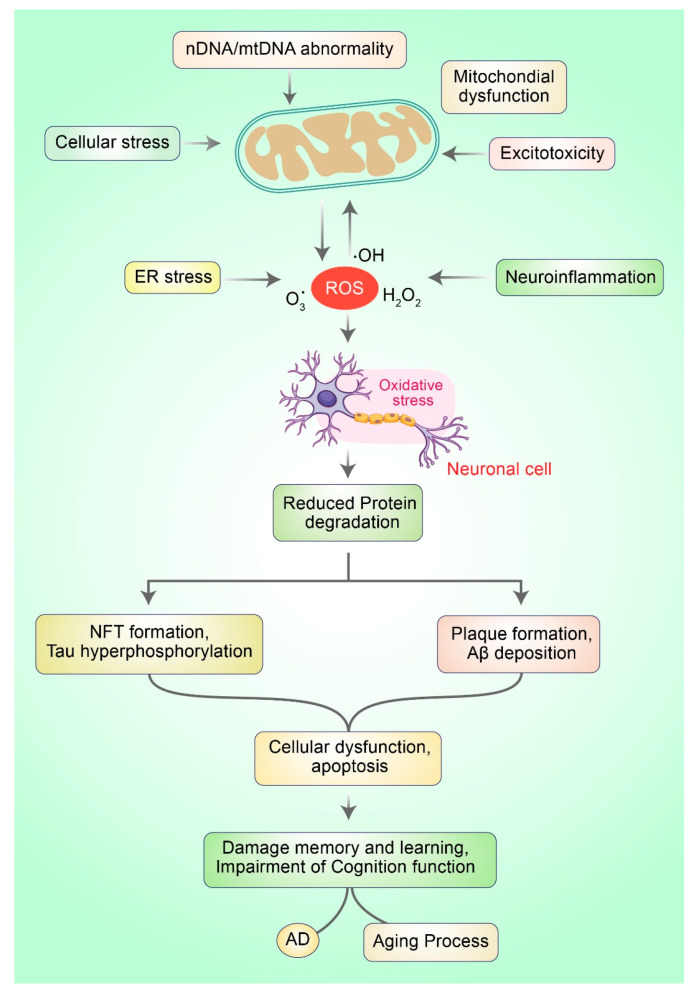
Mitochondrial dysfunction and oxidative stress in neurons lead to the development of AD. Typically, ROS are produced via numerous mechanisms such as ER stress, mitochondrial dysfunction, neuroinflammation, and excitotoxicity. Excessive ROS generation leads to oxi-dative stress (OS), which is responsible for mitochondrial dysfunction. OS prevents the deg-radation of protein molecules and impairs the clearance of misfolded proteins, which subse-quently leads to protein aggregation causing neuronal death and AD.

**Figure 2 antioxidants-10-00023-f002:**
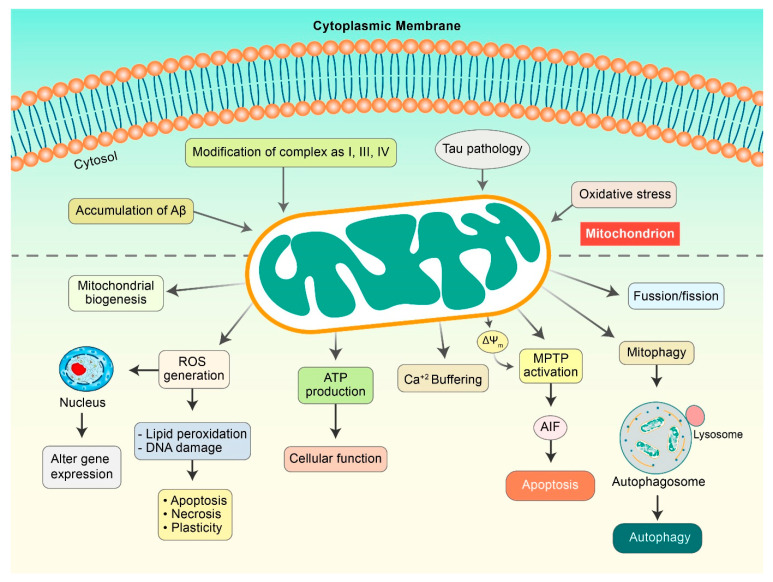
Mitochondrial dysfunction in AD pathogenesis. Aβ and Tau initiate mitochondrial dysfunctions that can result in the modulation of several factors. ROS is generated, which causes lipid peroxidation and DNA damage to initiate apoptosis. Damaged mitochondria demonstrate a decrease in mitochondrial membrane potential (ΔΨm) as a result of the activation of mitochondrial permeability transition pores (mPTPs), which release cytochrome c and apoptosis-inducing factor (AIF), and consequently, initiate apoptosis pathway. Aβ and pTau improve mitochondrial fission and mitophagy.

**Figure 3 antioxidants-10-00023-f003:**
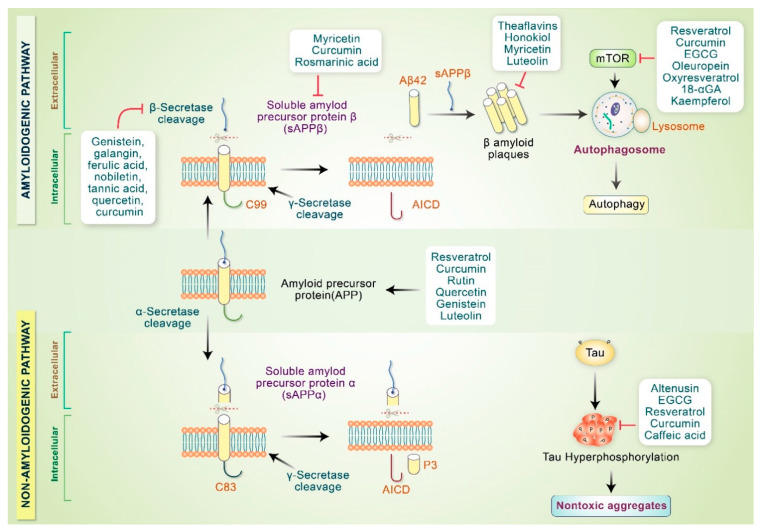
Phytochemicals modulate AD pathogenesis. Phytochemicals stimulate α-secretase activity or may hinder β-secretase activity that inhibits toxic oligomer production. Polyphenols and other compounds modify Aβ aggregates and convert them into nontoxic oligomers. Some phytochemicals inactivate mTOR and initiate the autophagy pathway. Polyphenols and other compounds prevent tau hyperphosphorylation and convert tau aggregates into nontoxic aggregates.

**Figure 4 antioxidants-10-00023-f004:**
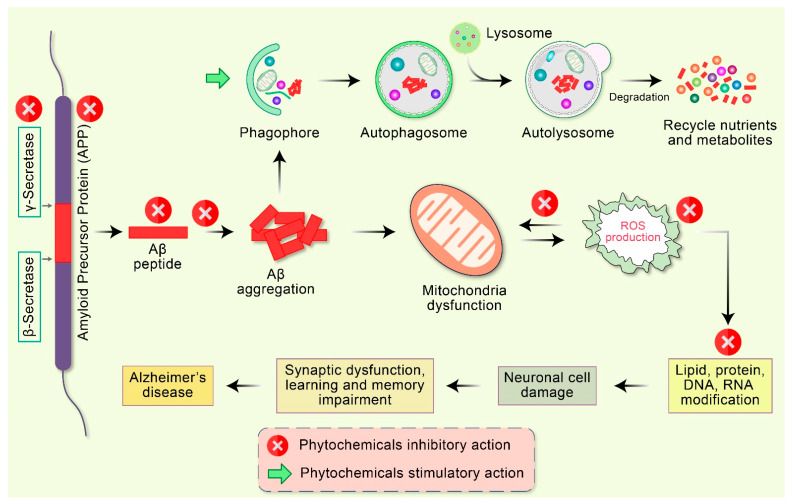
Emerging potential therapeutic targets of phytochemicals in mitochondrial dysfunction and AD pathogenesis. Abnormal APP is proteolyticlly cleaved by β- as well as γ-secretase leading to the accumulation of extracellular amyloid-β (Aβ). Deficient clearance of Aβ or Aβ production increases aggregation, leading to the accumulation of a diversity of Aβ assemblies. Accumulation of Aβ directly interrelates with mitochondria as well as ROS generation with different intracellular pathways. These oxidative stress reactions lead to the impairment of neuronal synapses and dendrite function with multifactorial mechanisms, in addition to neurological degeneration and dysregulation of synaptic function in the regions of the brain implicated in learning and memory impairment in AD. Additionally, Aβ aggregates are degraded by autophagy via the stimulatory action of phytochemicals.

**Table 1 antioxidants-10-00023-t001:** Different phytochemicals mitigating mitochondrial dysfunctions in AD pathology.

Phytochemicals	Experimental Model	Pathobiology	Research Outcomes	Molecular Signaling	References
Anthocyanins	APP Swedish K670N/M671L double mutation (APPswe)	Mitochondrial dysfunction and oxidative stress	Ameliorate mitochondrial dysfunction	Increased NADH levels	[34]
Resveratrol	Aβ-induced cytotoxicity in PC12 cells	Oxidative stress	Neuroprotection, Reduction of memory impairment	Reduced ROS, Induced SOD, PI3K, Akt	[49]
Tea polyphenols	SH-SY5Y cells	Oxidative stress	Neuroprotection	Keap1-Nrf2 signaling initiation	[41]
Sulfuretin	Aβ neurotoxicity in primary hippocampal neurons and SH-SY5Y cells	Oxidative stress	Neuroprotection	Activation of Nrf2/HO-1 and PI3K/Akt	[35]
Genistein	Transgenic APP/PS1 rat model of sporadic AD	Impairment of cognition, Increased β-amyloid and hyperphosphorylated tau protein	Improved learning and memory recognition, Inhibition of apoptosis and antioxidant functions	PPARγ-mediated increased release of ApoE, Autophagy activation and reduction in protein aggregates.	[50,51]
Liquiritigenin	Aβ-induced SK-N-MC cells	Mitochondrial fragmentation	Inhibited mitochondrial fragmentation and cytotoxicity	Mediated by Mfn1, Mfn2, and Opa1 signaling	[42]
Kaempferol	Porcine embryos	Oxidative stress	Prevented mitochondrial membrane potential and ROS generation.	Induced autophagy	[52]
Curcumin	Sprague Dawley male rats	Cerebral Ischemia	Neuroprotection	Autophagy and PI3K/Akt/mTOR pathway	[46]
Epigallocatechin-3-gallate (EGCG)	Rat primary cortical neuron	Pathological tau species	Enhanced tau degradation in an Nrf2-dependent manner	Increase autophagy, tau clearance	[53]
Quercetin	H_2_O_2_-induced neurotoxicity in Sprague-Dawley rat	Oxidative stress	Neuroprotection	Increased Aβ clearance	[54]

**Table 2 antioxidants-10-00023-t002:** Phytochemicals and other chemicals used for mitochondrial-targeted therapies in AD models in preclinical and clinical studies.

Phytochemical/Drug Candidate	AD Model	Mitochondrial Effect	References
Melatonin	HEK293-APPswe AD model	Increase of mitochondrial biogenesis and mitochondrial membrane potential, Decrease of APP processing	[118]
Coenzyme Q10	TgP301S mice, M17 cell line treated with Aβ_1-42_ peptide, HUVEC cell line Aβ_25-35_ peptide-treated	Decrease of ROS levels, Reduction of the accumulation of Aβ peptide, mtΔΨ protection, Promotion of ETC	[119,120]
Astaxanthin	Mouse hippocampal neurons treated by Aβ_1-42_ oligomers	Reduction of mitochondrial Production of H_2_O_2_	[121]
Resveratrol	APP/PSEN1 mice	Activation of mitophagy, Reduction of ROS accumulation	[122]
Pioglitazone	APP/PSEN1 mice	Reduction of Aβ1-42 level, Restoration of mitochondrial function	[123]
Dimebon	Mild-to-moderate AD patients	Improvement of cognition and memory function	[124]
Oxaloacetate (OAA)	AD cultured cells and mice	Activation of mitochondrial biogenesis	[125]
2-deoxyglucose	Adult rats treated with Aβ peptides	Increase in mitochondrial biogenesis, Reduction of mitochondrial stress	[126]
Curcumin	APP/PSEN1 mice, APP751SL mice	ROS reduction, Increase in synaptic function	[109]
Epigallocatechin-3-gallate (EGCG)	APP/PSEN1 mice	Restoration of mitochondria respiratory rates, Reduction of ROS and Aβ	[127]
Catalase	MCAT/APP mice	Reduction of oxidative damage, Aβ, BACE1 activity, and APP processing	[128]
α-lipoic acid	AD patients	Increase in cognition function, Protection against Aβ toxicity	[129]
N-Acetyl-cysteine (NAC)	A double-blind AD patient	Improvement of cognitive and behavioral functions	[130]
Quercetin	APP/PSEN1 mice	Improvement of ΔΨ, Prevent intrinsic apoptosis	[54]
G. biloba	Older adults and AD patients	Prevention of cognition and memory decline	[131]
SkQ1	OXYS rats	ROS reduction, COX increase	[132]
SS31	APP mouse model (Tg2576)	Decrease in Aβ production and dysfunction, Stimulation of mitochondrial biogenesis and	[133]
Ketones	3xTgAD)	Enhancement of mitochondrial functions and dynamics	[134,135]
Rapamycin	Aβ treated PC12 cell line	Increase in mitophagy	[136]
Red ginseng (RG)	5XFAD mice	Amelioration of Aβ deposition, Increase in mitochondrial biogenesis	[137]
Thiosemicarbazones	AD model of SK-N-MC neuroepithelioma cells	Inhibition of Aβ deposit formation, Reduction in ROS levels	[138]

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
