# Peer review of "Potential Therapeutic Role of Phytochemicals to Mitigate Mitochondrial Dysfunctions in Alzheimer’s Disease"

_antioxidants, 2020, doi:10.3390/antiox10010023_

Round 1

Reviewer 1 Report

This is a nice comprehensive review regarding mitochondria and oxidative stress in AD. The figures and tables are quite nice. However, some of the english writing is quite confusing and needs improvement. I would recommend the authors consult a colleague adept at written scientific english or utilize a service. Once the writing is improved, this will be a nice and useful manuscript ready for publication.

Author Response

This is a nice comprehensive review regarding mitochondria and oxidative stress in AD. The figures and tables are quite nice. However, some of the english writing is quite confusing and needs improvement. I would recommend the authors consult a colleague adept at written scientific english or utilize a service. Once the writing is improved, this will be a nice and useful manuscript ready for publication.

>>Response: First of all, we would like to express our sincere gratitude for the time and effort the reviewer had put into reviewing our manuscript. We massively revised and improved the quality of our manuscript by the professional English language editors (Company name: Editage, Ref#: MDATA_5).

Reviewer 2 Report

This review by Rahman et al. discusses the role of mitochondrial dysfunction and radical oxygen species in Alzheimer’s disease (AD) and the potential of phytochemicals as a potential therapeutic approach through their modulation of mitochondrial function. The manuscript is difficult to read due to English language issues (grammar, word choice, etc.) and editing for these issues is strongly recommended prior to acceptance. The authors provide a comprehensive catalog of phytochemicals that have been shown to protect cells from Abeta neurotoxicity and/or mitochondrial dysfunction and provide information on the likely pathways involve. It is not always clear in the text or table whether studies were performed in live animal models (i.e., analyzing brains) or cells cultured from them. It would benefit the paper to include when cells analyzed are from humans vs. animals, especially since mouse and rat models are not typically strong models of AD/do not replicate all the cellular and tissue effects.

Other things to consider:

Do phytochemicals pass the blood brain barrier?

Is there any supporting data from human population studies?

Lines 255-256 it is not clear what this means: “… when bound to the hydrophobic site of the amyloid pentamer” (when what is bound?).

Author Response

This review by Rahman et al. discusses the role of mitochondrial dysfunction and radical oxygen species in Alzheimer’s disease (AD) and the potential of phytochemicals as a potential therapeutic approach through their modulation of mitochondrial function. The manuscript is difficult to read due to English language issues (grammar, word choice, etc.) and editing for these issues is strongly recommended prior to acceptance. The authors provide a comprehensive catalog of phytochemicals that have been shown to protect cells from Abeta neurotoxicity and/or mitochondrial dysfunction and provide information on the likely pathways involve. It is not always clear in the text or table whether studies were performed in live animal models (i.e., analyzing brains) or cells cultured from them. It would benefit the paper to include when cells analyzed are from humans vs. animals, especially since mouse and rat models are not typically strong models of AD/do not replicate all the cellular and tissue effects.

>>Response: First of all, we would like to express our sincere gratitude for the time and effort the reviewer had put into reviewing our manuscript. We massively revised and improved the quality of our manuscript by the professional English language editors (Company name: Editage, Ref#: MDATA_5).

There are lots of evidences which supports beneficial role of phytochemicals have analyzed from humans and animals model. In the text, we mentioned several animal models are involved to use phytochemical as a potential treatment in AD. Here, we also added some phytochemicals which has a role in prevention of dysfunction of humans and animals studies. (page 6 line 154-159; page 15 line 349-355)

Other things to consider:

Do phytochemicals pass the blood brain barrier?

>>Response: We added several information in phytochemicals which have been found to penetrate blood brain barrier such as, dietary (poly)phenols, resveratrol, curcumin, gintonin, kaempferol, etc. (Page 7 line 172-173; page 8 line 178; line 190-191; line 194-198)

Is there any supporting data from human population studies?

>>Response: Phytochemicals, as plant components with discrete bio-activities towards animal biochemistry and metabolism are being widely examined for their ability to provide health benefits. Although, there is no enough evidences to study in human population. In our study, we hypothesis that human population trials are necessary to translate the existing findings into clinical use. Therefore, understanding the advanced pathobiology of AD and the pharmacological mechanism of phytochemical-based therapy may offer an emerging novel neuroprotective approach for AD in the future.

Lines 255-256 it is not clear what this means: “… when bound to the hydrophobic site of the amyloid pentamer” (when what is bound?).

>>Response: We checked the reference and modified the sentence. (page 12 line 282-284)